# Where to Look for a Remedy? Burnout Syndrome and its Associations with Coping and Job Satisfaction in Critical Care Nurses—A Cross-Sectional Study

**DOI:** 10.3390/ijerph18084390

**Published:** 2021-04-20

**Authors:** Adriano Friganović, Polona Selič

**Affiliations:** 1University Hospital Centre Zagreb, Kispaticeva 12, 10000 Zagreb, Croatia; adriano@hdmsarist.hr; 2University of Applied Health Sciences, Mlinarska cesta 38, 10000 Zagreb, Croatia; 3Department of Family Medicine, Faculty of Medicine, University of Ljubljana, Poljanski nasip 58, 1000 Ljubljana, Slovenia

**Keywords:** burnout, coping mechanisms, critical care, job satisfaction, nurse

## Abstract

Background: Burnout is a psychological, work-related syndrome associated with long-term exposure to emotional and interpersonal stressors in the workplace. Burnout syndrome in nurses is often caused by an imbalance between work requirements and preparation and fitness for work, a lack of control, insufficient performance recognition and a prolonged exposure to stress. Aim: The aims of this study were to explore the associations between levels of burnout syndrome, coping mechanisms and job satisfaction in critical care nurses in multivariate modelling process. A specific aim was also to explore whether coping and job satisfaction in critical care nurses are gender related. Methods: A cross-sectional multicentre study was conducted in a convenience sample of 620 critical care nurses from five university hospitals in Croatia in 2017. The data were collected using the Maslach Burnout Inventory and the Ways of Coping and Job Satisfaction Scale together with the nurses’ demographic profiles and were analysed using a multivariable model. Results: The results showed no significant association between gender, coping mechanisms and job satisfaction. However, significant negative associations between burnout and job satisfaction (OR = 0.01, 95%CI = 0.00–0.02, *p* < 0.001) and positive association between burnout and passive coping (OR = 9.93, 95%CI = 4.01–24.61, *p* < 0.001) were found. Conclusion: The association between job satisfaction and burnout in nurses urges hospital management teams to consider actions focused on job satisfaction, probably modifications of the work environment. Given that passive coping may increase the incidence of burnout, it is recommendable for active coping to be implemented in nurses’ training programmes as an essential element of capacity building aimed at reducing the incidence of burnout in nurses.

## 1. Introduction

Stress usually plays a very important role in the creation of psychological discomfort, disorders related to behaviour and difficulties in social adaptation [1]; such strain often leads to burnout, especially in healthcare workers. Burnout has been defined as a multidimensional concept which includes immense emotional exhaustion (EE) and depersonalization (DP) and a low sense of personal accomplishment (PA) [2]; it is known to be a job-related condition, associated with job satisfaction, as well as with coping mechanisms [3]. A key aspect of the burnout syndrome is increased feelings of emotional exhaustion; as emotional resources are depleted, nurses feel they are no longer able to give themselves at a psychological level [3]. In contemporary health psychology, coping is recognized as a highly important topic [4] in managing stress, in order to regulate a person’s emotional response [5]. Two Australian authors explored the possible “buffering effects” of using humour in coping with stress and the effect of job satisfaction on the stress-mood relationship [5]. Situational factors, such as situational demands of specific stressful encounters, are usually significant determinants of coping strategies and perceptions of stress [5]. Even more important is the perception and subjective interpretation of the stressor and how a person reacts to it [6], e.g., seeking support (instrumental or emotional) was found to be the most frequently used coping strategy [7]. Reducing the incidence of burnout by utilizing effective coping strategies has been shown to be an efficient measure, increasing job satisfaction in healthcare workers [8]. In the highly stressful and demanding environment of critical care, nurses’ evidence suggests that they experience high stress and burnout and low job satisfaction [9], which calls for further exploration. Guillermo et al. did a large literature review, 56 studies included, and found significant relation between depersonalization and gender [10]. This study aimed to discover whether coping mechanisms, with use of Folkman Lazarus Ways of Coping questionnaire, have a strong association with incidence of burnout [10]. The study also aimed to confirm differences in the relationship between burnout and gender to set scientific background for further research.

The critical care environment is highly demanding and stressful, and high levels of stress in nurses can cause depression and anxiety associated with burnout, as well as decreased job satisfaction and an increased intent to leave nursing practice [11,12,13]. Nurse burnout syndrome usually has financial repercussions for the healthcare system [14,15]. Nurses must recognize stress as part of their jobs, and support services can suggest improved coping strategies [7]. The concept of coping has been described as the set of strategies utilized by individuals in order to adapt themselves to adverse or stressful events [12]. The concept of burnout was examined in the context of human services, such as health care, social work, psychotherapy and teaching, and the main definition describes burnout as a syndrome of emotional exhaustion, depersonalization, and reduced personal accomplishment that can occur among individuals who work with people in some capacity“ [3]. A hospital, as an organization, should provide support for nurses as individuals, and professional^,^ as well as support services [7]. Personality traits, coping behavior and burnout are highly associated [16], adaptive personality traits are significantly positively associated with active coping styles, but these associations have not been fully explained, especially burnout [16]. Efforts towards the reduction of burnout should be based on a clear insight into the mechanisms through which personality traits and coping strategies exert their effects [14]. Coping processes that generate and sustain positive emotions tend to be different from those that regulate negative emotions; coping processes associated with positive emotions seem to be primarily appraisal-based [6].

Strategies used to cope with stressful situations at work may have important implications for well-being in the workplace, for example job satisfaction [14]. Job satisfaction is defined as a pleasurable or positive emotional state, resulting from the assessment of job experiences by using instruments, e.g., Job Satisfaction Scale [17,18,19]. Many theories concerning the causes of job satisfaction have been described, and it has been correlated with a number of outcome variables such as life satisfaction, job performance and withdrawal behaviours [16]. At the global level, the healthcare environment is developing more and more, and there is a need for a general understanding about fundamental work attitudes such as job satisfaction [17]. Life satisfaction and burnout were found to influence nurses’ confidence and preparedness for writing prescriptions and referrals for diagnostic tests [20]. Job satisfaction is poorly included as part of an organization’s key values, basic beliefs, core competencies or guiding principles, and the topic is not given much direct exposure in popular business books [16]. Job complexity definitely has a direct relationship with job satisfaction^3^. Burnout is a job-related condition and is associated with job satisfaction and with the degree of providing quality of care [19]. The highest level of job satisfaction has been identified in primary care nurses and the lowest in operating room nurses [21]. High levels of burnout are more common among physicians and nurses compared to other professions and associated with external factors such as: high workload and ineffective interpersonal relationships, e.g., with colleagues at work and family members [22]. Bartosiewicz and Januszewicz found the highest level of burnout to be related with psychophysical exhaustion [23].

A literature review found studies conducted on stress and coping in Croatia [15], but only one Croatian study included nurses as participants [24]; Mesar et al. aimed to determine the prevalence and intensity of stress and to identify the most intense stressors in two groups of nurses from two different surgical wards at University Hospital Dubrava [24]. A significant degree of stress among the respondents from both groups was verified, and the researcher identified staff issues and work overload as stressors with the highest intensity [25]. Most of the authors use Masclach Burnout Inventory (MBI) for burnout assessment, yet there are other instruments used in research, e.g., Professional Quality of Life Scale (ProQOL), the Spielberger State Trait Anxiety Inventory, the Copenhagen Burnout Inventory, Occupational Stressors Inventory, Moral Distress Scale-Revised, Nurse Stress Thermometer [26]. Critical care nurses have specific work conditions, surrounded by highly complicated equipment and requiring specialized training and education [27]. These specific conditions render them vulnerable to burnout and this study aimed to help reduce the incidence of burnout by exploring the variables that may lead to it [25]. Occupational variables, e.g., having workplace commitment, work freedom, possibilities for development at work, influence, and meaning of work, followed by psychological variables, e.g., being less extroverted and sociable, were identified as the most important in the burnout onset [28]. 

## 2. Research Aim and Questions

The aims of this study were to explore the associations between levels of burnout syndrome, coping mechanisms and job satisfaction in critical care nurses in multivariate modelling process. The research hypotheses for the study were:Coping and job satisfaction in critical care nurses are not gender related;Coping mechanisms and job satisfaction are associated with burnout syndrome.

## 3. Method

### 3.1. Study Design

A cross-sectional multicentre research design was used, and the study was carried out between April and September 2017.

### 3.2. Settings and Participants

A convenience sampling method was used, and the target population was critical care nurses employed in the Intensive Care Units (ICUs) of five Croatian university hospitals. The inclusion criteria for this study were more than six months of work experience in those who volunteered to participate. Exclusion criteria were work experience less than six months because of their short period of exposure to work-related stressors and the mentoring process still going on, namely first six months in ICU nurses do not act independently but work exclusively under the supervision of senior nurses. The period of the first six months was therefore considered as transitional and orientation learning period and the researcher decision was to exclude this group. Of the recruited critical care nurses, 80 rejected to participate; response rate was 88.57%.

In the Republic of Croatia, there are no official data on the number and gender structure of critical care nurses. To overcome this, a short survey was carried out and a total sum of critical care nurses and male population was calculated based on the answers provided by telephone. For the year of data collection, the approximate number was 3500 critical care nurses, of which approximately 13.0% were male nurses. It is therefore safe to conclude that the sample gender structure of this study reflects the gender structure of the population of critical care nurses in Croatia.

### 3.3. Instruments

Burnout was conceptualized as a continuous variable, ranging from low to high degrees of experienced feeling [3]. The Masclach Burnout Inventory (MBI) was used to assess the three components of burnout syndrome [3]; it consists of 22 items with three subscales: EE, DP and reduced PA. The items are rated using a 7-point, fully anchored scale (ranging from 0 “*never*” to 6 “*every day*”). The levels (scores) were considered high if they were in the upper third of the normative distribution, moderate if they were in the middle third and low if they were in the lower third, as suggested by Maslach et al. [27]. In this study, the MBI showed good internal consistency reliability with Cronbach’s alpha, ranging from 0.74 to 0.90 for each subscale which correlate with similar research [2].

*The Ways of Coping (WOC)* was used to assess what coping strategies people use to deal with the internal and/or external demands of specific stressful encounters [16]. The WOC is a 66-item questionnaire with eight coping strategies [29]: confrontive coping (CC), distancing (D), self-controlling (SC), seeking social support (SS), accepting responsibility (AR), escape-avoidance (EA), planful problem solving (PP) and positive reappraisal (PR). The response to each item is rated according to a 4-point Likert scale (0 = does not apply and/or not used; 1 = used somewhat, 2 = used quite a bit, 3 = used a great deal) marked on the participants’ thoughts and actions in dealing with stressful events. In this sample, all the WOC scales showed moderate reliability of internal consistency (Cronbach’s alpha between 0.6 and 0.7).

The *Job Satisfaction Scale* (JSS) was used to assess overall job satisfaction; it consists of five items rated from 1 to 5 (1 =strongly disagree; 5 = strongly agree) [16]. The five items are: *I feel fairly satisfied with my present job*; *Most days I am enthusiastic about my work*; *Each day at work seems like it never end*; *I find real enjoyment in my work; I consider my job to be rather unpleasant* [16]. The internal consistency reliability for JSS was Cronbach α < 0.80 *The Ways of Coping* (WOC) was used to assess what coping strategies people use to deal with the internal and/or external demands of specific stressful encounters [16].

### 3.4. Data Collection Procedures

Data collection was carried out between April and September 2017. Instruments were validated and adapted in Croatia in previous research [24,25,29,30]. Nurses were recruited directly by the researcher or with the help of head nurses of ICUs. Data collection was based on paper-pencil type of questionnaires. Ethics approvals were obtained from the Ethical Committees of all five University hospitals prior data collection (Zagreb University Hospital, 8.1-16/179-2, 21 November 2016; Sestre Milosrdnice University Hospital, EP-18818/16-2, 28 November 2016; Merkur University Hospital, 0311-12251, 8 December 2016; Sveti Duh University Hospital, 01-1916, 1 June 2017; Dubrava University Hospital, EP 17-05-2017, 17 May 2017). The study complied with the scientific principles of the Declaration of Helsinki. The purpose and nature of the study was explained to the nurses. They were assured that confidentiality would be maintained and that they were free to withdraw from the study at any time. The nurses were warned that the survey was not completely anonymous but that the researchers would provide anonymization for every participant by giving codes to each participant. The nurses were asked to sign a consent form and to complete the questionnaires in their own time and return the completed form to the researcher.

### 3.5. Data Analyses

Several independent variables were included in the regression modelling; therefore, in the first step the total sample size was divided by the variance inflation factor (VIF) to address the multicollinearity. The VIF, which depends on the squared multiple correlation coefficient (R^2^) relating a specific predictor of interest to the remaining predictors, was calculated according to the instructions in Hsieh [31]. We applied ordinal logistic regression and calculated R^2^ for JSS and WOC (F1 active coping, F2 passive coping) and used the maximum value obtained (R^2^ = 0.354). The effective sample size was reduced to 400 cases (total sample size of 620 divided by the VIF = 1/(1 − R^2^) = 1.55). Following Hsieh [31], the effective total of 400 participants was calculated to have more than 95% power to detect a significant association for logistic regression (using an alpha of 0.05, a medium odds ratio of about 2.5 to 1 and an event proportion of at least 5%) [32]. Data analyses were carried out using IBM SPSS Statistics for Windows, version 22.0 (IBM Corp., Armonk, NY, USA). Descriptive statistics (frequency, percentage, mean, standard deviation) were used to summarize the main characteristics of the sample. The internal consistency reliability (Cronbach Alpha) was calculated for each instrument.

For the WOC, we calculated the principal factor analysis with varimax rotation, the Kaiser-Meyer-Olkin measure of sampling adequacy gave a high value (KMO = 0.876), and the Bartlett’s test of sphericity was statistically significant (Chi-square = 1831.903, df = 28, *p* < 0.001), indicating the suitability of a factor analysis. Eight WOC dimensions were calculated as mean scores (the total raw scores were divided by the number of items in the scale) [29]. F1 active coping included five WOC dimensions (CC, SS, AR, PP and PR) which were again calculated as mean scores across all the scales [29]. F2 passive coping included three WOC dimensions (D, SC and EA). This calculation procedure was suggested by Folkman and Lazarus [29]. The relative F1 and F2 mean scores were rounded back to the original scale from 0 (*not used*) to 3 (*used a great deal*) to demonstrate a more common presentation of the results. By the rotated solution, F1 explained 25.9% of the initial variance and F2 explained 24.5% of the initial variance; both factors cumulatively explained 50.4% of the initial variance. The internal consistency reliability for active coping (F1) was 0.80 and for passive coping (F2) was 0.75.

Differences in JSS and WOC by gender were calculated using Chi-square test. Gender, age, coping and job satisfaction were selected as independent variables in the ordinal logistic regression modelling, with burnout (each dimension of MBI) being the dependent variable. Significance was set at *p* < 0.05.

### 3.6. Ethical Considerations

The study procedures were in accordance with the Declaration of Helsinki and the study was approved by the Ethical Committees of the five University Hospitals where the study was conducted. All the nurses gave written consent for their participation. They had the possibility of withdrawing or interrupting their participation at any time.

## 4. Results

### Demographic Profile of Participants

The sample included 620 critical care nurses: 544 women (87.7%) and 76 men (12.3%). The participants were aged 33.5 ± 9.5 years, in a range from 19 to 62 years; the female nursing staff were on average aged 26–35 years (*n* = 241 (38.9%)). Work experience was 10.3 ± 7.7 years in a range from 1 to 38 years; for most of the nurses, less than five years of work experience (*n* = 244 (39.4%)) was reported. The most common working department was general surgical ICU (*n* = 225 (36.3%)), at UHC Zagreb (*n* = 321 (51.8%)). There were 318 (51.3%) nurses with education at a vocational school level, and 248 with a bachelor’s degree (40.0%). More demographic characteristics are presented in Table 1.

In this sample, 137 (22.1%) of ICU nurses had a high level of EE and 49 (7.9%) of DP; 214 (34.5%) had a low level of PA, and 72 (11.6%) had burnout according to the MBI_tot_. Gender was not related to job satisfaction (Chi-square test *p* < 0.443). Active coping used *somewhat* was identified in 340 (62.5%) female and 48 (63.2%) male. Gender was also not related to coping mechanisms (active coping *p* < 0.927 and passive coping *p* < 0.144)

Gender was not related to job satisfaction (*p* < 0.443) or to coping mechanisms (active coping *p* < 0.927 and passive coping *p* < 0.144). Active coping used somewhat was identified in 340 (62.5%) women and 48 (63.2%) men. When participating nurses reported high burnout, passive coping was reportedly *used quite a bit* in a greater proportion (34.7%), but otherwise there were 11.6% nurses with low and 23.2% with medium burnout identified. Job satisfaction assessed as *neutral*, *satisfied* and *very satisfied* JSS was associated with a lower level of burnout (Table 2).

When the ordinal logistic regression model of associations with EE was performed, it explained 30% of the variance (Nagelkerke R^2^ = 0.302). In the model, a higher EE was negatively associated with job satisfaction assessed as *neutral* (OR = 0.33, 95%CI = 0.14–0.80, *p* = 0.014), *satisfied* (OR = 0.06, 95%CI = 0.03–0.15, *p* < 0.001) and *very satisfied* (OR = 0.01, 95%CI = 0.00–0.03, *p* < 0.001). A higher EE was associated with passive coping described as *used somewhat* (OR = 2.04, 95%CI = 1.02–4.06, *p* = 0.043) and *used quite a bit* (OR = 3.43, 95%CI = 1.54–7.65, *p* = 0.003) (Table 3).

In the second modelling process, logistic regression was performed with DP as the dependent variable. This time, only 17% of the variance was explained (Nagelkerke R^2^ = 0.176). Higher level of DP was negatively associated with job satisfaction assessed as *neutral* (OR = 0.18, 95%CI = 0.08–0.40, *p* < 0.001), *satisfied* (OR = 0.07, 95%CI = 0.03–0.16, *p* < 0.001) and *very satisfied* (OR = 0.04, 95%CI = 0.02–0.11, *p* < 0.001). Higher level of DP was associated with passive coping described as *used somewhat* (OR = 2.99, 95%CI = 1.25–7.17, *p* = 0.014) and *used quite a bit* (OR = 7.74, 95%CI = 2.94–20.39, *p* < 0.001) and male gender (OR = 2.03, 95%CI = 1.20–3.42, *p* = 0.008) (Table 3).

The third logistic regression model was performed with PA as the dependent variable, and 19% of its variance was explained (Nagelkerke R^2^ = 0.186). Higher level of PA was positively associated with job satisfaction assessed as *satisfied* (OR = 4.04, 95%CI = 1.81–9.03, *p* = 0.001) and *very satisfied* (OR = 10.40, 95%CI = 1.81–9.03, *p* < 0.001), and bachelor’s degree (OR = 1.40, 95%CI = 1.01–1.96, *p* = 0.045) and Master’s level of education (OR = 2.39, 95%CI = 1.34–4.26, *p* = 0.003). Lower level of PA was associated with 5–10 years of work experience (OR = 0.56, 95%CI = 0.33–0.95, *p* = 0.032) (Table 3).

Finally, MBI_tot_ was used as a dependent variable in the logistic regression. Higher level of MBI_tot_ was negatively associated with job satisfaction assessed as *neutral* (OR = 0.22, 95%CI = 0.10–0.52, *p* = 0.001), *satisfied* (OR = 0.04, 95%CI = 0.02–0.09, *p* < 0.001) and *very satisfied* (OR = 0.01, 95%CI = 0.00–0.02, *p* < 0.001). Higher level of MBI_tot_ was associated with passive coping described as *used somewhat* (OR = 3.08, 95%CI = 1.39–6.83, *p* = 0.006), *used quite a bit* (OR = 9.93, 95%CI = 4.01–24.61, *p* < 0.001) and five to ten years of work experience (OR = 1.99, 95%CI = 1.10–3.60, *p* = 0.024). Using MBI_tot_ as a measure of burnout in the modelling process was shown to be useful, and the largest part of the variance was explained, nearly 36% (Nagelkerke R^2^ = 0.359) (Table 3).

## 5. Discussion

The aim of our study was to determine associations between burnout, job satisfaction and coping mechanisms. Aside from this, we wanted to explore the relationships between gender, job satisfaction and coping in ICU nurses. The study showed no relationship between gender and job satisfaction and coping; we only found a relationship between gender and DP, which can be explained by the fact that nursing is a traditionally female profession (see gender structure in the sample, Table 3) [27]. However, only 17% of the variance was explained in that model, and we believe that except for gender and other variables in the model, there should have been other, probably organizational and workload-related, variables to explain the DP in ICU nurses.

However, the results of this study showed an association of passive coping and job satisfaction to the burnout dimensions EE and DP, as well as MBI_tot_ (Table 3). Specifically, we found research from authors in Jordan who found that coping strategies could be a moderating factor which could improve compassion and job satisfaction in critical care nurses [33]. They also stated that the focus should be on increasing and learning coping strategies and providing a supportive work environment [33], which is concordant with our results that higher EE was negatively associated with job satisfaction assessed as *neutral*, *satisfied* and *very satisfied* (Table 3). Our results could be used when planning the organization of educational activities concerning active coping in order to reduce burnout, as the authors from Jordan suggested [6], which could be a feasible way of addressing the high burnout results. Given that burnout and coping were associated (Table 2), results of this study may be considered in concordance with other studies [34]. 

Chinese researchers conducted a study with a cross-sectional design in tertiary hospitals with the aim of examining the association between work stress and coping strategies [35] and reported that active coping strategies have an impact in reducing the negative effect of work stress on job performance; on the other hand, the researchers reported that passive coping strategies increased the negative effects, which is also concordant with our results (Table 2). Another group of Chinese authors [35] stated that active coping was positively related to resource and environmental problems, and passive coping was positively related to workload and time pressure and to interpersonal relationship and management issues [36], which is more or less similar to the findings of our study e.g., higher levels of MBI_tot_ were associated with passive coping described as *used somewhat* (Table 2). The implications of the Chinese study on nursing practice are that nurse managers have the ability to reduce the use of passive coping and to prevent burnout by teaching nurses active coping strategies, which may be also beneficial for Croatian ICU nurses according to the results of this study (Table 2).

Our results (Table 2) show that the use of active coping strategies to reduce burnout was aligned with a study hypothesis that stress severity and four maladaptive coping behaviours are related to incidence of burnout [37]. Welbourne et al. concluded that coping strategies were associated with job satisfaction and higher burnout [38], similarly to the results of this study, i.e., that *somewhat used* passive coping and *quite a bit used* passive coping are associated with lower level of job satisfaction (Table 2).

In the PA dimension we did not find a significant association with coping strategies, but we did reveal an association with higher educational level and work experience. These findings could be used as advice for nurse management, similarly to the reference that managers need to have the ability to ensure and promote higher education for nurses [38].

In Serbia, researchers conducted a study on the role of personality dimensions and coping strategies in physicians [30] and confirmed a correlation between escape-avoidance (passive coping) with high results of DP. This is quite well aligned to our findings that *somewhat used* passive coping and *quite a bit used* passive coping were associated with higher DP. The results of our study with regard to job satisfaction being associated with a lower degree of burnout MBI_tot_ (Table 2) show that hospitals should create a healthy environment (job satisfaction, safety in the workplace, staffing, autonomy of practice) for healthcare workers. Adawarkwah et al. in Germany explored work satisfaction in general practitioners practicing in rural areas, with results important for clinical management [39]; they found that professionals with a higher EE need different intervention from respondents with a high level of DP [39,40,41]. Morsiani et al. stated that the main attributes of a transformational leadership style, e.g., respect, caring for others and appreciation, showed higher job satisfaction results [42], which could also be used as motivation to hospital administrators in Croatia, given that higher job satisfaction was found to be associated with reduced burnout (Table 2); the finding was also confirmed by many other authors [40,41,42]. Labrague et al., in their integrative review, showed the importance of enhancing social support and promoting job control and stressed that hospital administrators have a significant role in promoting supportive structures; results of our study also show a way to reduce burnout [43]. The professional benefit of this study includes the potential to introduce and adopt functional coping strategies. Given that active coping may decrease the incidence of burnout, training and capacity building should acknowledge this finding.

Considering the results of our study and other research and literature reviews, there is still a need for deeper research into aspects of job satisfaction in critical care nurses using a different study design. Aside from this, some leadership changes could be made to obtain better satisfaction with work-related interpersonal relationships, with a special emphasis on developing constructive coping skills mechanisms. The characteristics of supervision, ICU policy and work procedures, as well as job control and job attitudes, need to be studied thoroughly in the future. Performing desirable work tasks appeared to be of similar importance to achieving additional training.

## 6. Study Limitations

One of the first limitations is the reduced anonymity of the study. These results form part of a larger multicentre mixed study with both quantitative and qualitative approaches. The qualitative part was undertaken by nurses with high burnout, so they had to be identified based on results of the MBI and the study could not be completely anonymous. It would be interesting to conduct research on a representative sample which provided complete anonymity.

A cross-sectional survey design is inherently limited and, together with reliance on self-reported data, raises questions about the potential for method variance to account for our findings. However, the phenomenon being studied could only be assessed by asking patients to report their experience or perception. Prospective studies using clear diagnostic criteria and measures, as well as in-depth qualitative studies, would be beneficial for extending and deepening our understanding of bio-psycho-social patterns in nurses with burnout. We believe that further research should also focus on a longer period of time, in order to get more detailed characteristics and better grounds for preventive action planning at the social level and also in the field of education.

## 7. Conclusions

This study brought new knowledge and gave new direction for curriculum programmes in nursing education, which should include knowledge of and skills in burnout and coping strategies. Since level of education was found to be associated with a lower incidence of personal accomplishment, nursing management should encourage nurses to take part in further/continuous education.

Finding that demographic data seemingly had no associations with the level of burnout points to the fact that all healthcare professionals are susceptible to burnout. It is therefore important to raise awareness of the possible problems deriving from stress and burnout, since burnout is connected to a greater economic burden due to a large amount of sick leave, more frequent changing of workplace, lower working efficiency and early retirement.

Strengthening the job-satisfaction related factors may reduce the impact of burnout in critical care nurses. With regard to the association between job satisfaction and incidence of burnout, hospital management should consider improvements in the work environment. Preventive strategies should not be only for critical care nurses but for all nurses in healthcare institutions, as well as other healthcare professionals, e.g., physicians, physiotherapists and occupational therapists.

## Figures and Tables

**Table 1 ijerph-18-04390-t001:** Demographic characteristics of the sample (*n* = 620).

	*n*	%
Gender		
Male	76	12.3
Female	544	87.7
Age in years		
18–25	141	22.7
26–35	241	38.9
36–45	161	26.0
>45	77	12.4
Education		
Vocational school	318	51.3
Bachelor’s degree	248	40.0
Master’s degree	54	8.7
Marital status		
Single	279	45.0
Married	316	51.0
Divorced or widowed	25	4.0
Work experience in years		
<5	244	39.4
5–10	99	16.0
11–15	90	14.5
16–20	70	11.3
>20	117	18.9
Working department		
Cardiac-surgical ICU	80	12.9
Neuro-surgical ICU	58	9.4
Paediatric and neonatal ICU	73	11.8
Medical ICU	80	12.9
Surgical ICU	225	36.3
Coronary ICU	72	11.6
Neurological ICU	32	5.2

**Table 2 ijerph-18-04390-t002:** Burnout (MBI_TOT_) according to demographic characteristics, job satisfaction and coping.

Variable	Burnout			*p* *
	Low *n* = 354 (%)	Medium *n* = 194 (%)	High *n* = 72 (%)	
Gender				0.669
Male	40 (11.3)	27 (13.9)	9 (12.5)	
Female	314 (88.7)	167 (86.1)	63 (87.5)	
Age in years				0.270
18–25	83 (23.4)	45 (23.2)	13 (18.1)	
26–35	134 (37.9)	79 (40.7)	28 (38.9)	
36–45	86 (24.3)	49 (25.3)	26 (36.1)	
>45	51 (14.4)	21 (10.8)	5 (6.9)	
Education				0.556
Vocational school	180 (50.8)	99 (51.0)	39 (54.2)	
Bachelor’s degree	138 (39.0)	83 (42.8)	27 (37.5)	
Master’s degree	36 (10.2)	12 (6.2)	6 (8.3)	
Marital status				0.772
Single	162 (45.8)	84 (43.3)	33 (45.8)	
Married	177 (50.0)	101 (52.1)	38 (52.8)	
Divorced or widowed	15 (4.2)	9 (4.6)	1 (1.5)	
Work experience in years				0.111
<5	148 (41.8)	73 (37.6)	23 (31.9)	
5–10	44 (12.4)	38 (19.6)	17 (23.6)	
11–15	52 (14.7)	29 (14.9)	9 (12.5)	
16–20	36 (10.2)	22 (11.3)	12 (16.7)	
>20	74 (20.9)	32 (16.5)	11 (15.3)	
JSS				<**0.001**
Dissatisfied	3 (0.8)	8 (4.1)	17 (23.6)	
Neutral	37 (10.5)	68 (35.1)	36 (50.0)	
Satisfied	238 (67.2)	115 (59.3)	16 (22.2)	
Very satisfied	76 (21.5)	3 (1.5)	3 (4.2)	
Active coping				0.951
Not used	12 (3.4)	6 (3.1)	3 (4.2)	
Used somewhat	224 (63.3)	118 (60.8)	46 (63.9)	
Used quite a bit	118 (33.3)	70 (36.1)	23 (31.9)	
Passive coping				<**0.001**
Not used	39 (11.0)	12 (6.2)	4 (5.6)	
Used somewhat	274 (77.4)	137 (70.6)	43 (59.7)	
Used quite a bit	41 (11.6)	45 (23.2)	25 (34.7)	

* Chi-square test. Bolded are statistically significant associations

**Table 3 ijerph-18-04390-t003:** Logistic regression model of variables associated with EE, DP, PA and MBI_tot_.

	OR (95% CI)	*p*	OR (95% CI)	*p*	OR (95% CI)	*p*	OR (95% CI)	*p*
JSS								
Unsatisfied	1.00 (reference)		1.00 (reference)		1.00 (reference)		1.00 (reference)	
Neutral	0.33 (0.14, 0.80)	**0.014**	0.18 (0.08, 0.40)	**<0.001**	1.35 (0.58, 3.12)	0.487	0.22 (0.10, 0.52)	**0.001**
Satisfied	0.06 (0.03, 0.15)	**<0.001**	0.07 (0.03, 0.16)	**<0.001**	4.04 (1.81, 9.03)	**0.001**	0.04 (0.02, 0.09)	**<0.001**
Very satisfied	0.01 (0.00, 0.03)	**<0.001**	0.04 (0.02, 0.11)	**<0.001**	10.40 (4.23, 25.62)	**<0.001**	0.01 (0.00, 0.02)	**<0.001**
Active coping								
Not used	1.00 (reference)		1.00 (reference)		1.00 (reference)		1.00 (reference)	
Used somewhat	0.41 (0.15, 1.16)	0.093	0.59 (0.18, 1.96)	0.387	0.91 (0.35, 2.37)	0.843	0.48 (0.16, 1.47)	0.199
Used quite a bit	0.51 (0.17, 1.52)	0.227	0.53 (0.15, 1.86)	0.317	1.38 (0.50, 3.83)	0.531	0.39 (0.12, 1.27)	0.116
Passive coping								
Not used	1.00 (reference)		1.00 (reference)		1.00 (reference)		1.00 (reference)	
Used somewhat	2.04 (1.02, 4.06)	**0.043**	2.99 (1.25, 7.17)	**0.014**	0.94 (0.50, 1.75)	0.841	3.08 (1.39, 6.83)	**0.006**
Used quite a bit	3.43 (1.54, 7.65)	**0.003**	7.74 (2.94, 20.39)	**<0.001**	0.69 (0.33, 1.45)	0.329	9.93 (4.01, 24.61)	**<0.001**
Gender								
Female	1.00 (reference)		1.00 (reference)		1.00 (reference)		1.00 (reference)	
Male	1.54 (0.94, 2.53)	0.087	2.03 (1.20, 3.42)	**0.008**	1.20 (0.74, 1.93)	0.456	1.32 (0.78, 2.24)	0.300
Age in years								
18–25	1.00 (reference)		1.00 (reference)		1.00 (reference)		1.00 (reference)	
26–35	1.02 (0.59, 1.76)	0.944	0.88 (0.48, 1.59)	0.663	1.29 (0.78, 2.14)	0.316	0.96 (0.53, 1.74)	0.899
36–45	1.14 (0.54, 2.37)	0.736	0.97 (0.43, 2.18)	0.947	1.09 (0.54, 2.20)	0.801	1.37 (0.62, 3.01)	0.434
>45	1.59 (0.65, 3.87)	0.308	1.16 (0.43, 3.10)	0.770	1.39 (0.60, 3.24)	0.442	1.14 (0.43, 3.00)	0.790
Education								
Vocational school	1.00 (reference)		1.00 (reference)		1.00 (reference)		1.00 (reference)	
Bachelor’s degree	1.24 (0.87, 1.75)	0.233	0.97 (0.66, 1.43)	0.863	1.40 (1.01, 1.96)	**0.045**	0.92 (0.63, 1.35)	0.679
Master’s degree	0.83 (0.45, 1.54)	0.556	0.77 (0.39, 1.53)	0.455	2.39 (1.34, 4.26)	**0.003**	0.54 (0.27, 1.08)	0.082
Work experience in years								
<5	1.00 (reference)		1.00 (reference)		1.00 (reference)		1.00 (reference)	
5–10	1.29 (0.74, 2.25)	0.374	1.09 (0.59, 2.01)	0.785	0.56 (0.33, 0.95)	**0.032**	1.99 (1.10, 3.60)	**0.024**
11–15	1.13 (0.62, 2.06)	0.687	1.09 (0.56, 2.13)	0.799	1.06 (0.60, 1.87)	0.847	1.11 (0.57, 2.14)	0.762
16–20	1.33 (0.64, 2.76)	0.439	0.89 (0.39, 2.00)	0.768	1.23 (0.62, 2.47)	0.557	1.17 (0.53, 2.55)	0.699
>20	1.00 (0.47, 2.11)	0.990	0.93 (0.40, 2.17)	0.873	1.26 (0.62, 2.56)	0.533	0.88 (0.39, 1.98)	0.751
	Nagelkerke R^2^ = 0.302, OR: odds ratio, 95% CI: 95% confidence interval		Nagelkerke R^2^ = 0.176, OR: odds ratio, 95% CI: 95% confidence interval		Nagelkerke R^2^ = 0.186, OR: odds ratio, 95% CI: 95% confidence interval		Nagelkerke R^2^ = 0.359, OR: odds ratio, 95% CI: 95% confidence interval	

Bolded are statistically significant associations.

## Data Availability

The data presented in this study are available on request from the corresponding author. The data are not publicly available due to policy of institutions which gave ethical approval to the study.

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
