# Peer review of "Where to Look for a Remedy? Burnout Syndrome and its Associations with Coping and Job Satisfaction in Critical Care Nurses—A Cross-Sectional Study"

_ijerph, 2021, doi:10.3390/ijerph18084390_

Round 1

Reviewer 1 Report

Dear authors,

Your manuscript is interesting but I need you to answer some questions:

INTRODUCTION

  • The introduction should not be to have subsections. The subsection called "Introduction" and "background" should be a single section called "introduction".
  • Page 2, line 53: Who is Guillermo et al? This author does not come in the references. This author does not correspond to reference number 10. Therefore, the rest of the references should be reviewed.

METHOD

3.2. Settings and participants:

  • The authors consider that the experience of less than 6 months does not generate stress. Inexperience in a service such as the ICU can be a confounding variable for this research. Considering 6 months as a learning period is not correct. They are contract nurses, not nursing students. Why? You must justify your answer.
  • The authors must include the response rate of the participants in the study.

3.4. Data collection procedures

CONCLUSIONS

  • The authors have not written "conclusions" but "implications for clinical practice".
  • Authors should explain clinical practice implications at the end of the "discussion".

REFERENCES

  • Many bibliographies are obsolete and some citations are incomplete. The bibliographic citations used are more than 5 years old (58,33%). The authors must update and arrange the bibliography.
  • There is an updated bibliography of original and meta-analytic articles that should be cited, among others.
  • Some references that have errors. The authors should review this section.

Author Response

Answers to the Reviewers` Comments

Thank you for the valuable comments. We applied all reviver’s suggestions and explained changes and modifications below. Throughout the manuscript we used red font colour to make it easier tracking changes; hopefully it is acceptable for reviewers. We changed the brackets according the IJERPH citation style.

REVIEWER #1

INTRODUCTION

  1. The introduction should not be to have subsections. The subsection called "Introduction" and "background" shouldbe a single sectioncalled "introduction".

A: Your suggestion was followed/applied at page 2 line 60. Subsection ''Background'' was removed to be part of undivided section called ''Introduction''.

Line 60: Background removed.

  1. Who is Guillermo et al? This author does not come in the references. This author does not correspond to reference number 10. Therefore, the rest of the references should be reviewed.

A: Your suggestion was followed/applied page 12, lines 438-440. The reference was not in the list by mistake. It was added and all references checked according to this suggestion.

Lines 438-440: Guilermo, A.C.D.F.; Ortega, E.; Ramirez-Baena, L.; Fuente Solana, E.; Vargas, C.; Gomez-Urquiza, J.L. Gender, Marital Status, and Children as Risk Factors for Burnout in Nurses: A Meta-Analytic Study. Int J Environ Res Public Health.2018, 15, 2102.doi: 10.3390/ijerph15102102.

METHODS

3.2. Settings and participants:

  1. The authors consider that the experience of less than 6 months does not generate stress. Inexperience in a servicesuch as the ICU canbe a confounding variable for this research. Considering 6 months as a learning period is not correct. They are contract nurses, not nursing students. Why? You must justify your answer.

A: Your suggestion was followed/applied at page 3, lines 134 - 137. Explanation is the mentoring processes established in ICU in Croatia.

Lines 134-137: Exclusion criteria were work experience less than six month because of their short period of exposure to work-related stressors and the mentoring process still going on, namely first six month in ICU nurses don`t act indipendenty but work exclusively under the supervision of senior nurses. Period of first six months was therefore considered as transitional and orientation learning period and researcher decision was to exclude this group.

  1. The authors must include the response rate of the participants in the study.

A: Your suggestion was followed/applied at page 3, line 139. Response rate has been added to the text.

Line 139: Response rate was 88.57%.

  1. 4. Data collection procedures

A: Your suggestion was followed/applied at page 4 lines 176-178. Data collection procedure is described in more detail in the Data collection section.    

Lines 176-178: Nurses were recrouted directly by researcher or with help of head nurses of ICU’s. Data collection based on paper-pencil type of questionnaires. 

CONCLUSIONS

  1. The authors have not written "conclusions" but "implications for clinical practice".

A:  Your suggestion was followed/applied at page 10 lines 351-354, page 11 lines 386-391. Practice implications were removed from conclusion (see below). Conclusions section was modified.

 Lines 351-354: The professional benefit of this study includes the potential to introduce and adopt functional coping strategies. Given that active coping may decrease the incidence of burnout, training and capacity building should acknowledge this finding.

Lines 386-391: Finding that demographic data seemingly had no associations with the level of burnout points to the fact that all healthcare professionals are susceptible to burnout. It is therefore important to raise awareness of the possible problems deriving from stress and burnout, since burnout is connected to a greater economic burden due to a large amount of sick leave, more frequent changing of workplace, lower working efficiency, and early retirement.

  1. Authors should explain clinical practice implications at the end of the "discussion".

A: Your suggestion was followed/applied at page 10 lines 379-362. Practice implications were moved from conclusion to the end of discussion section.      

Lines 344-347: The professional benefit of this study includes the potential to introduce and adopt functional coping strategies. Given that active coping may decrease the incidence of burnout, training and capacity building should acknowledge this finding.

REFERENCES

  1. Many bibliographies are obsolete and some citations are incomplete. The bibliographic citations used are more than 5 years old (58,33%). The authorsmust update and arrange the bibliography.

A: Your suggestion was followed/applied at page 12 lines 458-459, lines 464-465. Bibliography was updated and rearranged, therefore all references are presented with red fonts.

Lines 458-459: Bartosiewicz, A.; Łuszczki, E.; Dereń, K. Personalized Nursing: How Life Satisfaction and Occupational  Burnout Influence New Competences of Polish Nurses. J. Pers. Med. 2020, 10, 48. https://doi.org/10.3390/jpm10020048

Lines 459-460: Bartosiewicz, A.; Januszewicz, P. Readiness of Polish Nurses for Prescribing and the Level of  Professional Burnout. Int. J. Environ. Res. Public Health 2019, 16, 35. https://doi.org/10.3390/ijerph16010035

  1. There is an updated bibliography of original and meta-analytic articles that should be cited, among others.

A: Your suggestion was followed/applied at page 12 lines 462-463, lines 471-473, lines 476-478, page 13 lines 486-487. Original and meta – analytic articles were cited.

Lines 462-463: Garcia, C.d.L.; Abreu, L.C.d.; Ramos, J.L.S.; Castro, C.F.D.d.; Smiderle, F.R.N.; Santos, J.A.d.; Bezerra, I.M.P. Influence of Burnout on Patient Safety: Systematic Review and Meta-Analysis. Medicina 2019, 55, 553. https://doi.org/10.3390/medicina55090553

Lines 471-473: Molina-Praena, J.; Ramirez-Baena, L.; Gómez-Urquiza, J.L.; Cañadas, G.R.; De la Fuente, E.I.; Cañadas-De la Fuente, G.A. Levels of Burnout and Risk Factors in Medical Area Nurses: A Meta-Analytic Study. Int. J. Environ. Res. Public Health 2018, 15, 2800. https://doi.org/10.3390/ijerph15122800

Lines 476-478: Gómez-Urquiza, J.L.; Albendín-García, L.; Velando-Soriano, A.; Ortega-Campos, E.; Ramírez-Baena, L.; Membrive-Jiménez, M.J.; Suleiman-Martos, N. Burnout in Palliative Care Nurses, Prevalence and Risk Factors: A Systematic Review with Meta-Analysis. Int. J. Environ. Res. Public Health 2020, 17, 7672. https://doi.org/10.3390/ijerph17207672

Lines 486-487: Sok, S.; Sim, H.; Han, B.; Park, S.J. Burnout and Related Factors of Nurses Caring for DNR Patients in Intensive Care Units, South Korea. Int. J. Environ. Res. Public Health 2020, 17, 8899. https://doi.org/10.3390/ijerph17238899

  1. Some references that have errors. The authors should review.

A: Your suggestion was followed/applied at page 11 – 13 lines 422 - 507.The reference list was thoroughly corrected.

Reviewer 2 Report

The manuscript needs to be improved before publication.

Details in the attachment.

Author Response

Answers to the Reviewers` Comments

Thank you for the valuable comments. We applied all reviver’s suggestions and explained changes and modifications below. Throughout the manuscript we used red font colour to make it easier tracking changes; hopefully it is acceptable for reviewers. We changed the brackets according the IJERPH citation style.

REVIEWER #2

Comments for authors Thank you for the opportunity to review the article “Where to look for a remedy? Burnout Syndrome and its Associations with Coping and Job Satisfaction in Critical Care Nurses – A Cross-Sectional Study”.

  1. The work deals with a very important topic but before publication the manuscript needs thorough improvement: The aim of the study, both in the abstract and in the main documents, is not very precise - please correct it.

A: Your suggestion was followed/applied at page 1 lines 14-15, page 3 lines 120-122. We tried to be more specific with research aim.

Lines 14-15, lines 120-122: The aims of this study were to explore the associations between levels of burnout syndrome, coping mechanisms and job satisfaction in critical care nurses in multivariate modelling process.

  1. In the introduction, it is worth adding information about the current scale of occupational burnout among nurses in Europe and in the world.

A: Your suggestion was followed/applied at page 3 lines 107 -111.

Lines 107-111: Most of the authors use Masclach Burnout Inventory (MBI) for evaluating levels of burnout, but there is other instruments used by researchers such as the Professional Quality of Life Scale (ProQOL),  the Spielberger State Trait Anxiety Inventory, the Copenhagen Burnout Inventory, Occupational Stressors Inventory, Moral Distress Scale-Revised, Nurse Stress Thermometer, etc [26].

  1. Please describe the recruitment rules and sampling methods.

A: Your suggestion was followed/applied at page 4 lines 176-178. We tried to better describe recruitment rules and sampling methods.

Lines 176-178: Nurses were recrouted directly by researcher or with help of head nurses of ICU’s. Data collection based on paper-pencil type of questionnaires.

  1. The results are not clearly presented. It is worth presenting the level of occupational burnout, the way of coping and job satisfaction in the surveyed group.

A: Your suggestion was followed/applied. Another Table, i.e. Table 2 (Table 2. Burnout (MBITOT) according to demographic characteristics, job satisfaction and coping) was added (see Line 246) and Table 3 renumbered accordingly.

With this additional presentation, there were also some parts of the text added:

Lines 248-254: Gender was not related to job satisfaction (p˂0.443), or to coping mechanisms (active coping p˂0.927 and passive coping p˂0.144). Active coping used somewhat was identified in 340 (62.5%) women and 48 (63.2%) men. When participating nurses reported high burnout, passive coping was reportedly used quite a bit in a greater proportion (34.7%), but otherwise there were 11.6% nurses with low and 23.2% with medium burnout identified. Job satisfaction assessed as neutral, satisfied and very satisfied JSS was associated with a lower level of burnout (Table 2).

Lines 357-362: Aside from this, some leadership changes could be made to obtain better satisfaction with work-related interpersonal relationships, with a special emphasis on developing constructive coping skills mechanisms. The characteristics of supervision, ICU policy and work procedures, as well as job control and job attitudes, need to be studied thoroughly in the future. Performing desirable work tasks appeared to be of similar importance to achieving additional training.

Lines 386-391: Finding that demographic data seemingly had no associations with the level of burnout points to the fact that all healthcare professionals are susceptible to burnout. It is therefore important to raise awareness of the possible problems deriving from stress and burnout, since burnout is connected to a greater economic burden due to a large amount of sick leave, more frequent changing of workplace, lower working efficiency, and early retirement.

  1. Please add the number and date of the Ethics Committee.

A: Your suggestion was followed/applied at 4 lines 179-183. Number and date of the Ethics Committee has been added to main text.

Lines 179-183: (Zagreb University Hospital, 8.1-16/179-2, 21 November 2016; SestreMilosrdnice University Hospital, EP-18818/16-2, 28 November 2016; Merkur University Hospital, 0311-12251, 8 December 2016; Sveti Duh University Hospital, 01-1916, 1 June 2017; Dubrava University Hospital, EP 17-05-2017, 17 May 2017).

  1. In the discussion, it is worth adding examples of occupational burnout studies among nurses, with other than MBI questionnaire e.g.:

https://www.mdpi.com/2075-4426/10/2/48/htm

https://www.mdpi.com/1660-4601/16/1/35/htm

https://www.mdpi.com/2077- 0383/8/1/92/htm

A: Your suggestion was followed/applied page 2 lines 88-90, lines 99-100.These references have been added to the main text.

Line 88-90: Life satisfaction and burnout were found to influence nurses` confidence and preparedness for writing prescriptions and referrals for diagnostic tests [20]. 

Line 99-100: Bartosiewicz and Januszewicz found the highest level of burnout to be related with psychophysical exhaustion [23].

Third reference couldnt be find, the link was incorrect.

  1. References and abstracts need to be improved in accordance with the instructions for authors: Please see: https://www.mdpi.com/journal/ijerph/instructions.

A: Your suggestion was followed/applied at page 11-13 lines 422-507. The references has been checked and corrected.

Lines 415-502: The references were corrected. 

Round 2

Reviewer 1 Report

Dear authors,

Thanks for your reply. The explanations of the authors are satisfactory. The paper has greatly improved its quality.

You just have to correct a small mistake:

INTRODUCTION

  • The reference is wrong. It is not Guillermo et al, but Cañadas-De la Fuente et al. You must write the reference well.

REFERENCES

  • Some references that have errors. Reference 10 is wrong. You must correct it.

Congratulations on your work.

Best regards

Author Response

Distinguished Reviewer #1,

Thank you for your assistance and additional valuable observations. We applied your suggestions and explained all changes below.

Throughout the manuscript we used red font colour to make it easier tracking changes.

INTRODUCTION

  • The reference is wrong. It is not Guillermo et al, but Cañadas-De la Fuente et al. You must write the reference well.

A: Your suggestion was followed/applied on page 12, lines 438-440. The reference was corrected, we apologise for this overseen mistake.

Lines 438-440: Candance-De la Fuente G.A.; Ortega, E.; Ramirez-Baena, L.; Fuente Solana, E.; Vargas, C.; Gomez-Urquiza, J.L. Gender, Marital Status, and Children as Risk Factors for Burnout in Nurses: A Meta-Analytic Study. Int J Environ Res Public Health.2018, 15, 2102.doi: 10.3390/ijerph15102102.

REFERENCES

Some references that have errors. Reference 10 is wrong. You must correct it.

 A: Your suggestion was followed/applied page 12, lines 438-440. The reference corrected.

Lines 438-440: Candance-De la Fuente G.A.; Ortega, E.; Ramirez-Baena, L.; Fuente Solana, E.; Vargas, C.; Gomez-Urquiza, J.L. Gender, Marital Status, and Children as Risk Factors for Burnout in Nurses: A Meta-Analytic Study. Int J Environ Res Public Health.2018, 15, 2102.doi: 10.3390/ijerph15102102.

A: Your suggestion was followed/applied through pages 11 – 13, lines 422 - 507.The reference list was thoroughly inspected.

Reviewer 2 Report

Accept in present form

Author Response

Distinguished Reviewer #2,

Thank you for your encouragement.